# Dietary and Plasma Carboxymethyl Lysine and Tumor Necrosis Factor-α as Mediators of Body Mass Index and Waist Circumference among Women in Indonesia

**DOI:** 10.3390/nu11123057

**Published:** 2019-12-14

**Authors:** Patricia Budihartanti Liman, Rina Agustina, Ratna Djuwita, Jahja Umar, Inge Permadhi, Adi Hidayat, Edith J.M. Feskens, Murdani Abdullah

**Affiliations:** 1Department of Nutrition, Faculty of Medicine, Universitas Indonesia - Dr. Cipto Mangunkusumo General Hospital, Jakarta 10430, Indonesia; yuttika2000@gmail.com (P.B.L.); ingepermadhi@yahoo.com (I.P.); 2Department of Nutrition, Faculty of Medicine, Universitas Trisakti, Jakarta 11440, Indonesia; 3Human Nutrition Research Center, Indonesian Medical Education and Research Institute (IMERI), Faculty of Medicine, Universitas Indonesia, Jakarta 10430, Indonesia; djuwita257@gmail.com (R.D.); murdani08@gmail.com (M.A.); 4Southeast Asian Ministers of Education Organization Regional Center for Food and Nutrition (SEAMEO RECFON)—Pusat Kajian Gizi Regional Universitas Indonesia, Jakarta 10430, Indonesia; 5Department of Epidemiology, School of Public Health, Universitas Indonesia, Depok 16424, Indonesia; 6Department of Psychology, Faculty of Psychology, Universitas Islam Negeri Syarif Hidayatullah, Jakarta 15419, Indonesia; umarindo@me.com; 7Department of Nutrition, Faculty of Public Health, Universitas Andalas, Padang 25163, Indonesia; 8Department of Community Medicine, Faculty of Medicine, Universitas Trisakti, Jakarta 11440, Indonesia; hidayat.adi@trisakti.ac.id; 9Division of Human Nutrition, and Health, Wageningen University, 6708 WE, Wageningen, The Netherlands; edith.feskens@wur.nl; 10Department of Internal Medicine, Faculty of Medicine, Universitas Indonesia-Dr. Cipto Mangunkusumo General Hospital, Jakarta 10430, Indonesia

**Keywords:** body mass index, waist circumference, carboxymethyl lysine, central obesity, Indonesian women, tumor necrosis factor-α

## Abstract

Dietary and plasma carboxymethyl lysine (dCML, pCML) and plasma tumor necrosis factor-α (pTNF-α) may be associated with obesity in affluent society. However, evidence in women from low-middle income countries with predominantly traditional diets is lacking. We investigated the mediator effects of dCML, pCML and pTNF-α on body mass index (BMI) and waist circumference (WC) among Indonesian women. A cross-sectional study was conducted among 235 non-diabetic, non-anemic and non-smoking women aged 19–50 years from selected mountainous and coastal areas of West Sumatra and West Java. Dietary CML, pCML, pTNF-α were obtained from 2 × 24-h recalls, ultra-performance liquid chromatography-tandem mass spectrometry and enzyme-linked immunosorbent assay, respectively. Between-group differences were analyzed by the Chi-square or Mann-Whitney test and mediator effects by Structural Equation Modeling. The medians and interquartile-ranges of dCML, pCML and pTNF-α were 2.2 (1.7–3.0) mg/day, 22.2 (17.2–28.2) ng/mL and 0.68 (0.52–1.00) IU/mL, respectively, and significantly higher in the WC ≥ 80 cm than in the WC < 80 cm group, but not in BMI ≥ 25 kg/m^2^ compared to BMI < 25 kg/m^2^ group. Plasma CML and pTNF-α were positively and directly related to WC (β = 0.21 [95% CI: 0.09, 0.33] and β = 0.23 [95% CI: 0.11, 0.35], respectively). Dietary CML that correlated with dry-heat processing and cereals as the highest contributions was positively related to WC (β = 0.33 [95% CI: 0.12, 0.83]). Ethnicity, level of education, intake of fat, and intake of energy contributed to dCML, pCML and pTNF-α, and subsequently affected WC, while only ethnicity contributed to BMI through dCML, pCML and pTNF-α (β = 0.07 [95% CI: 0.01, 0.14]). A positive direct effect of dCML on pCML and of pCML and pTNF-α on WC was seen among Indonesian women. Dietary CML seems to have several paths that indirectly influence the increases in WC if compared to BMI. Thus, intake of CML-rich foods should be reduced, or the foods consumed in moderate amounts to avoid the risk of central obesity in this population.

## 1. Introduction

The significant increase in the prevalence of obesity has become an important public health problem worldwide and in developing countries, including Indonesia [1,2]. According to the National basic health survey, the prevalence of obesity in Indonesia has increased from 14.8% in 2013 to 21.8% in 2018 [2], while the obesity burden is higher in women than in men [1,2]. The shift in dietary patterns from the traditional to a western diet, especially in urban areas, is one of the key drivers to the increased energy density and total calorie intake affecting the body weight [3]. In addition, the current unhealthy food environment has influenced the people’s food choices to an easy access of fast foods and processed foods [4,5,6]. Moreover, industrial foods for catering or other commercial purposes have been processed and modified for tastes, colors and smells to satisfy consumer preferences. Alteration in food processing will impact, not only on increasing energy-dense dietary intake, but may also increase the formation of advanced glycation end products (AGEs) in these foods from exogenous AGEs sources apart from smoking [7,8].

Exogenous AGEs in food contribute more to the AGEs in the blood when compared with endogenous AGEs [9] that are formed through the glycolytic and protein or lipid metabolic pathways. Previous studies showed that the formation of AGEs occurs spontaneously in vivo [10,11], and can be induced by hyperglycemic states [11,12]. The results of kinetic studies on AGE metabolism showed that 10% of AGEs in the food that we consume will be absorbed and only one-third of this amount will be excreted in the urine [8,13,14]. The remaining AGEs in the body will accumulate, leading to the risk of non-communicable diseases. An imbalance between the amounts of AGEs that are present in the body and of those that are excreted, can cause health problems [7,15].

The other factors influencing obesity are: level of education [16], ethnicity, age and physical activity [17]. These factors could have effects on AGEs [18,19,20,21] and further contribute to the risk of obesity and central obesity.

Carboxymethyl lysine (CML), a non-fluorescent protein adduct [22,23], is frequently used as a marker for the presence of AGEs [23,24,25]. Binding between AGEs and their receptors (RAGE) can cause cellular dysfunction by interfering with cellular communication, causing changes in protein structure and function, and mitochondrial dysfunction, leading to cell death [7,26,27]. Binding to RAGE can also increase reactive oxygen species (ROS), activate inflammatory signaling (tumor necrosis factor alpha, TNF-α), and induce insulin resistance, which is often associated with obesity [28].

Ethnic, cultural and residential diversity will affect food characteristics and individual dietary habits, which can then affect their health outcome. For example, two Indonesian ethnic groups, the Minangkabau and Sundanese, which are located in the West Sumatra and West Java provinces, respectively, have different types of cuisine. Minangkabau food is dominated by coconut milk, and is processed by boiling for long periods of time, while the Sundanese cuisine has more fresh vegetables, and dishes are usually cooked by frying [29,30]. Cooking at high temperatures, with low moisture content and long periods of time, is known to increase the formation of AGEs in foods. 

Although several studies have demonstrated a relationship between AGE markers and obesity in Caucasians [9,15,31,32], there are no reports of previous studies among Asians. The authors of the present study found only one study involving Asians, but these comprised only 4% of 130 subjects, and the recruitment was in New York City [31].

In the present study, we determined the associations of AGE markers such as dietary CML (dCML) with body mass index (BMI) and waist circumference (WC) as obesity and central obesity parameters, respectively, through the plasma CML (pCML), and plasma TNF-α (pTNF-α) parameters, in Indonesian women.

## 2. Materials and Methods

### 2.1. Subjects and Study Design

Subjects of this study were selected from the original cross-sectional study under the title “Association of intake and nutritional status with total microbiota and metabolic markers in Minangkabau and Sundanese women”. Minangkabau women were selected from the women living in districts representing mountainous areas (Tanah Datar) and coastal areas (Padang Pariaman) in West Sumatra Province. Sundanese women were selected from mountainous and coastal areas of the Tasikmalaya District in West Java Province. Apparently healthy women aged 19–50 years, having both parents of Minangkabau or Sundanese ethnicity, and having an active reproductive status, were admitted into the study (Figure 1).

The inclusion criteria of subjects were those who were not pregnant or lactating, not consuming alcohol and smoking, not having gastrointestinal disturbance for the last two days, having no malignancy, without a history of hospitalization in the last month, not taking antioxidant supplements in the last two weeks, not having diabetes, as known from the interview or from laboratory examination (HbA1c), and not having anemia (normal hemoglobin level), as described previously elsewhere by Stefani et al. [33].

The minimum number of 220 subjects (110 subjects for each ethnic group) was required for analyzing the mediator effect of dCML on obesity with 11 measured parameters (i.e., ethnicity, age, level of education, intake of fat, intake of energy, physical activity, dCML, pCML, pTNF-α, BMI and WC) using structural equation modeling (SEM). Bentler and Chow [34] showed that the minimum sample size can be determined by the ratio of the number of subjects to the number of measured variables, i.e., 5:1 to 10:1.

Of a total of 360 women, 267 subjects meeting the inclusion criteria were randomly selected with stratification by age and the proportion of obese and non-obese to obtain 120 individuals from each ethnic group. Five subjects were excluded because of missing laboratory data. Finally, 235 women (117 ethnic Minangkabau and 118 ethnic Sundanese) were included in the data analysis. All subjects signed written, informed consent, and this study was approved by the Health Research Ethics Committee, Faculty of Medicine, Universitas Indonesia-Dr. Cipto Mangunkusumo Hospital; and registered in www.clinicaltrials.gov under ID NCT03412617.

### 2.2. Dietary Carboxymethyl Lysine

First, lists of foods were obtained from 2-day, nonconsecutive 24-h dietary recalls, carried out on one working-day and one weekend-day by trained enumerators. The field-enumerators were selected through an academic qualification potential test and interviews. They had minimally an associate or bachelor degree in Nutrition/Public Health, and were experienced in conducting anthropometric examination and collecting 24-h food recall data. Training was given to the field-enumerators to standardize the procedure. The subjects were visited at home by the field-enumerators, then were asked to describe the preparation of their meals and to estimate the amount of foods that they had consumed on the day prior to the interview using household measures, aided by the food dish photographs from “Survey of Individual Food Consumption 2014” (“Survei Konsumsi Makanan Individu 2014”, SKMI-2014) [35]. This study analyzed dietary CML (dCML) using a database of the CML content of foods that had been developed, as described previously elsewhere by Liman et al. 2019 [36]. The CML values were the averages of 2-day recalls and stated as mg per 100 g edible part of food.

### 2.3. Anthropometric Assessments

Body weight, height and WC were obtained using calibrated digital scales for weight (SECA 876, Hamburg, Germany), ShorrBoard stadiometer (Weigh and Measure, LLC; Olney, MD, USA), and SECA cloth tape, respectively. Subjects were lightly clothed, without shoes and their accessories, before undergoing the measurement. Body weight was measured by asking the subjects to stand with both feet at the center of the scale. Height was measured with the ShorrBoard placed on the floor and against a flat, vertical surface. The subject was asked to stand relaxed, looking straight forward, with head, shoulder, buttocks and heels touching the board, the arms at the sides, and the legs straight. The headpiece of the measuring board was lowered slowly until it touched the top of the head, and was lowered further until the subject’s hair was depressed. The measurer’s eyes were at the same level as the headpiece. The site of the waist circumference measurement was along a line from the midpoint between the lowest rib and the iliac crest to the umbilicus. The subject stood with the abdomen relaxed, both arms at the sides and the feet close together. The subject was also recommended to breathe normally, and the measurement was performed without compressing the abdominal skin. Each measurement was repeated twice, and the average value was used. Weight, height and waist circumference were measured to the nearest 0.1 kg, 0.2 cm and 0.1 cm, respectively. The BMI was calculated from the ratio of body weight (kg) to height squared (m^2^). The categories of obesity according to the classification for the Asia Pacific region were non-obese (<25 kg/m^2^), and obese (≥25 kg/m^2^). Waist circumference was categorized as normal (<80 cm) and central obesity (≥80 cm).

### 2.4. Physical Activity

Physical activity was obtained using the short form of the international physical activity questionnaire (IPAQ) that was validated by Craig et al. in 2003 [37]. The physical activity categories were inactive physical activity: <600 MET and active physical activity: ≥600 MET.

### 2.5. Measurement of Plasma CML and Plasma TNF-α

The blood samples were drawn after an overnight fast of 10–12 h. Plasma CML was measured by ultra-performance liquid chromatography-tandem mass spectrometry (UPLC-MS/MS) in frozen ethylenediaminetetraacetic acid (EDTA)-plasma that had been thawed once and stored at −20 °C. LC-MS/MS analysis was performed at an internationally accredited and standardized mass spectrometry laboratory in Jakarta. The UPLC system was Agilent Liquid Chromatography (LC) System 1290 with Agilent Triple Quad (LC/MS) 6460 from Agilent Technologies, Santa Clara, CA, USA. The standard and internal standard were Nε-(1-Carboxymethyl)-L-Lysine from Chem Cruz, Santa Cruz Biotechnology, Dallas, TX, USA, catalog no: sc-212438 and Nε-(1-Carboxymethyl)-L-Lysine-(4,4,5,5-D4) from Cambridge Isotope Laboratories, Inc, Tewksbury, MA, USA, catalog no: DLM-4731-PK. Intra- and inter-day coefficients of variation (CV) were 2.9% and 10.2%, respectively.

Plasma TNF-α was measured by enzyme-linked immunosorbent assay (ELISA) in frozen EDTA- plasma using the Quantikine^®^ ELISA Human TNF-α HS kit (R & D Systems, Inc., Minneapolis, MN, USA), catalog no. HSTA00E. The measurement of pTNF-α was done in a private qualified research laboratory in Jakarta, Indonesia. The microplate reader was Bio-Rad Model 680 with Microplate Manager software (Bio-Rad Laboratories Inc., Yaherakles, CA, USA). The inter-assay coefficient of variation (CV) was 6.7%.

### 2.6. Statistical Analysis

The data was analyzed using SPSS program version 20.0 and Mplus program version 8.3, and yielded the following details: Categorical data were presented as proportion and continuous data as mean ± standard deviation (SE) for normally distributed data and median (25th percentile–75th percentile) for non-normally distributed data. For testing the normality of the data, the Kolmogorov-Smirnov test was used at the significance level of *p* < 0.05. Differences in mean or median values between groups were tested by the *t*-test or Mann-Whitney U test, respectively, depending on the normality distribution. Pearson’s correlation coefficient was calculated between two continuous variables. Structural equation modeling (SEM) was used to test the hypothesis of dCML as the mediator of BMI and WC. There are five successive stages in SEM: i.e., model specification, parameter identification, parameter estimation, assessment of model fit, and finally testing the hypothesis regarding parameters of the model. However, the model can be modified when the model fit is not achieved. The tests are two-sided for the *t*-test or Mann-Whitney U test, and one-sided for SEM, with the significance level of *p* < 0.05 and *p* < 0.025, respectively.

## 3. Results

The prevalence of obesity and central obesity was 48.5% (121/235) and 55.7% (131/235), respectively. Out of the 121 non-obese women, 32 (26.4%) women had a WC of more than 80 cm, and out of the 114 obese women, 15 (13.2%) women had a WC of less than 80 cm. The BMI had a positive correlation with WC in all the subjects (r = 0.817, *p* < 0.001). The median age of obese women was higher than that of non-obese women, but not statistically significantly different. The median age of central obese women was statistically significantly higher than that of non-central obese women (Table 1). The educational level and physical activity of non-obese and non-central obese women did not differ from that of obese and central obese women. Sundanese women had a higher prevalence of obesity and central obesity than Minangkabau women, the difference being statistically significant for central obesity.

Median dCML in all subjects was 2.2 mg/day (25th and 75th percentiles were 1.6 mg/day and 3.0 mg/day, respectively). Median pCML and pTNF-α were 22.2 ng/mL (22.2 and 28.2 ng/mL) and 0.68 IU/mL (0.52 and 1.00 IU/mL), respectively. Median dCML in obese women was higher than in non-obese women, but the difference was not statistically significant (Table 2), while the median dCML among central obese women (2.3 mg/day) was significantly higher than the median dCML among non-central obese women (2.1 mg/day). The median pCML and pTNF-α values among central obese women were higher than in non-central obese women (*p* = 0.026 and *p* < 0.001, respectively). 

The median pCML among central obese women was 23.2 ng/mL, and among non-central obese women, it was 21.0 ng/mL. The median pTNF-α of 0.76 IU/mL in women with central obesity was higher than that of non-central obesity women, with a median value of 0.62 IU/mL.

### SEM Pathways

In path analysis, it was found that the hypothesized model (Figure 2) fit the data well (RMSEA = 0.036), with the probability that RMSEA is smaller than 0.05 being 0.728. Therefore, the theoretical model regarding the role of CML as the mediator variable in influencing BMI and WC was accepted.

It was also found in this study that waist circumference was directly and positively related to pCML (β = 0.21, *p* < 0.025), pTNF-α (β = 0.23, *p* < 0.025) and intake of energy (β = 0.21, *p* < 0.025) (Table 3). A direct and positive effect of dCML on pCML (β = 1.44, *p* < 0.025) and of pCML on pTNF-α (β = 0.12, *p* < 0.025) was also seen. Waist circumference was positively related to dCML with sum of indirect effects β = 0.33, *p* < 0.025. In this case, there were two significant indirect paths from dCML to WC, the first path was through the mediators of dCML, pCML and pTNF-α (β = 0.04, *p* < 0.025), and the other path was through the mediators of dCML and pCML (β = 0.29, *p* < 0.025). In this model, it was found that dCML, pCML and pTNF-α played a role as mediators from ethnicity, intake of fat and intake of energy to WC (Table 4). Dietary CML, pCML and pTNF-α did not always give positive associations with WC, since there were some paths that showed a negative association with WC, depending on the other mediators and exogenous variables involved in the paths.

Ethnicity was positively related to WC with the sum of indirect effects β = 0.18, *p* < 0.025. There were four paths yielding a positive association between ethnicity and WC, namely through the mediators of (1) intake of energy, dCML and pCML (β = 0.02, *p* < 0.025), (2) dCML, pCML and pTNF-α (β = 0.01, *p* < 0.025), (3) dCML and pCML (β = 0.06, *p* < 0.025) and (4) through the mediator pTNF-α (β = 0.09, *p* < 0.025). We also found one other path yielding a negative association between ethnicity and WC, namely through the mediators of energy intake and pCML (β = −0.02, *p* < 0.025).

Plasma CML was directly and positively associated with dCML (β = 1.44, *p* < 0.025) but negatively associated with the intake of energy (β = −0.90, *p* < 0.025). A higher pTNF-α was seen in Sundanese ethnicity (β = 0.40, *p* < 0.025). Plasma CML was positively related to WC through the mediator pTNF-α (β = 0.03, *p* < 0.025).

Level of education was significantly associated with WC through four indirect paths. Two paths had a positive association with WC, namely through the mediators of fat intake, energy intake, dCML and pCML (β = 0.03, *p* < 0.025) and through the mediators of energy intake and pCML (β = 0.02, *p* < 0.025). The other two pathways of level of education had a negative association with WC, namely through the mediators of fat intake, energy intake and pCML (β = −0.03, *p* < 0.025) and through the mediators of energy intake, dCML and pCML (β = −0.02, *p* < 0.025).

Intake of fat was not directly related to WC (β = −0.14, *p* > 0.025), but significantly associated with WC through five indirect paths. Three paths had a positive association with WC, namely through the mediators of energy intake, dCML, pCML and pTNF-α (β = 0.02, *p* < 0.025), through mediators of energy intake, dCML and pCML (β = 0.14, *p* < 0.025), and through the mediator of energy intake (β = 0.17, *p* < 0.025). The other two pathways had a negative association with WC, namely through the intake of energy, pCML and pTNF-α (β = −0.02, *p* > 0.025) and through the mediators of energy intake and pCML (β = −0.15, *p* > 0.025).

Intake of energy was directly and positively related to WC (β = 0.21, *p* < 0.025). Intake of energy was also indirectly associated with WC, with two paths that were positively associated with WC, namely through dCML, pCML and pTNF-α (β = 0.02, *p* < 0.025), and through the mediators of dCML and pCML (β = 0.17, *p* < 0.025). The other two pathways yielding a negative association with WC, namely through the mediators of pCML and pTNF-α (β = −0.02, *p* < 0.025) and through the mediator of pCML (β = −0.18, *p* < 0.025).

However, BMI was found to be indirectly positively related only to ethnicity (β = 0.07, *p* < 0.025), but not to level of education, intake of fat, intake of energy, dCML, pCML and pTNF-α (*p* > 0.025). There were four pathways from ethnicity to BMI that involved dCML, namely through the mediators of energy intake, dCML, pCML and pTNF-α (β = 0.00, *p* > 0.025), through the intake of energy, dCML and pCML, (β = 0.01, *p* > 0.025), through dCML, pCML and pTNF-α (β = 0.00, *p* > 0.025), and through dCML and pCML (β = 0.03, *p* > 0.025).

In this study, physical activity was found to be negatively related to pCML (β = −0.05, *p* > 0.025), BMI (β = −0.07, *p* > 0.025), and WC (β = −0.06, *p* > 0.025), but the relationship was not statistically significant.

## 4. Discussion

The present study showed that the increment of one level of pCML and pTNF-α was associated with an increase in WC among Indonesian women of 0.21 and 0.23 points, respectively, but not with BMI. Among the 30 indirect pathways to WC in the model, there were twelve paths that involved dCML as a mediator, and most of the paths had a positive association with WC. Dietary CML as an independent variable was proven to have a positive indirect association with the greatest effect on WC, where a one-unit increment of dCML was associated with an increase of WC of 0.33 points. This showed that although ethnicity, percentage of fat intake, energy intake, or the level of education of the subjects affect the dCML, it seems that the amount of dCML itself has the major influence on obesity.

To minimize the over- and under-estimation of dCML, a specific food CML database for Indonesian foods was established that was compiled from published data on CML values measured by means of LC-MS/MS [36]. In calculating the values of CML for the individual foods, a well-ordered step-based approach was used, as was described before in Liman’s study. [36] Firstly, the CML values were most preferably obtained from the matched food items based on the similarity of names, the description, and of the processing methods. If a match was not found, the steps to estimate the CML value were (1) using recipes, (2) using similar foods in the same botanical or zoological genus or family with the same cooking method, and (3) lastly, using the CML food group values. Furthermore, an adjustment by protein content was done, which is essential to obtain a more accurate CML content. The food preparation methods, such as frying, boiling and grilling, were also considered in estimating the CML values. In order to minimize the subjects’ bias in dietary recall or the observation bias between interviewers, we trained the interviewers and used the book of food dish photographs [35] to provide visual aids in assessing the portion size consumed. All the aforementioned aspects confirmed that the dCML findings were true values and were not obtained by chance.

Dietary CML also played a role as a mediator from ethnicity to BMI, with the sum of indirect effects of β = 0.07, *p* < 0.025, but none of the specific paths separately yielded a significant effect on BMI. This indicates the possibility that it requires the simultaneous running of all specific paths to induce a positive effect of ethnicity on BMI. Further research is needed to confirm these findings. Dietary CML played a role as a mediator from ethnicity to WC (with a stronger effect on WC than on BMI), with sum of indirect effect β = 0.18, *p* < 0.025. In half of the specific paths from ethnicity to WC, we found that dCML played a role as a mediator, and all of the paths gave a positive and significant effect on WC. Interestingly, in a Western study, Uribarri et al. showed that CML did not differ among ethnicities [31], while in an Asian study [21] Ahmed et al. showed that the CML concentration was higher in the Malay ethnic group, compared to the Chinese and Indian ethnic groups. The present study also showed that the Sundanese ethnic group has higher dCML and pCML if compared to the Minangkabau ethnic group. This could mean that the dietary pattern in different Asian ethnicities is stronger than in Western countries; therefore, ethnicity should be considered as confounder if we want to observe the effect of CML in Asian subjects.

Median dCML in obese and central obese was 10% higher than in non-obese and non-central obese women, but a significant difference was found only between central obese and non-central obese. This indicated that a high intake of dCML may increase the risk of central obesity. In SEM, this concept was further strengthened with the existence of a number of significant paths that involved dCML as a mediator particularly toward WC.

In the guideline for the prevention of cardiovascular disease (CVD), as one of NCDs, a healthy lifestyle is recommended. This comprises the consumption of healthful foods, such as fruit, vegetables, fiber, fish, legumes and limitation in the consumption of sugar, trans fat, red meat and processed meat, the regular performance of moderate-intensity physical exercise and weight management in obese subjects [38,39]. However, in practice, this is not easy to apply, particularly in women, because of the strong influence of income, social role, educational level and ethnicity. Poor knowledge of nutrition and health, limitation in the purchase of healthful foods (which generally are more expensive), and lack of time for physical activity, are the inhibitory factors in the application of a healthy life style. Although some papers state that females seldom follow these recommendations in comparison to males, the mortality rate from CVD is higher in males, especially in younger age [40]. This may be caused by the protective effect of the estrogens [39,40] and the sex chromosome of females [41,42]. If the consumption of high dietary CML is considered as an unhealthy diet, then the present study supports the statement above, by showing that ethnicity, level of education, intake of fat and intake of energy, are related to dCML and are further associated with WC, as one of the CVD risk factors, while only ethnicity is related to dCML and associated with BMI.

A limitation of the current study was that no information was available about the duration of heating and the cooking temperature, which also influence CML formation. In obtaining dCML data, we used 2-day, non-repeated, 24-h food recall, which relies on the subjects’ memory, and may not reflect the usual intake.

The UPLC-MS/MS method with standard and internal standard was used in the measurement of pCML, and the intra- and inter-day coefficients of variation to maintain the accuracy and reproducibility of the measurements were calculated. This method was used by the previous studies as one of several methods for measuring CML in plasma or foods [43]. The ELISA method was chosen for measuring plasma TNF-α because it had been widely used in immunology-based research and clinical studies. In measuring the pTNF, an attempt to reduce the inter-assay coefficient of variation to less than 10% was made by carrying out the measurements simultaneously on the same day.

Mean pCML was positively associated with BMI and WC. As a major part of AGEs in the body [44], CML could modulate the pathogenesis of inflammation, oxidative stress, β-cell apoptosis, endothelial dysfunction, and insulin resistance, and lead to the pathogenesis of obesity and central obesity [28,45]. Uribarri et al. [31] showed that obese subjects with metabolic syndrome, of which one of the symptoms is WC > 88 cm in women, had a higher serum CML than obese subjects without metabolic syndrome. The present study also showed that the pCML had a stronger association with WC than with BMI. However, Gaens et al. [15] showed different results, i.e., that pCML levels in subjects with obesity based on BMI and WC were lower when compared with normal subjects.

TNF-α is a proinflammatory cytokine that has a role in lipid metabolism and insulin signaling. In obesity, inflammation could occur due to an accumulation of fat in the body as a result of several mechanisms. Excessive intake of food will lead to fat tissue expansion and adipocyte hypertrophy, resulting in local hypoxia and adipocyte apoptosis. This condition stimulates macrophage recruitment [46]. Adipocyte hypertrophy will lead to elevated levels of free fatty acids which further stimulate macrophages to produce TNF-α [47]. The present study demonstrated that pTNF-α has a stronger association with WC than with BMI (β = 023 and β = 0.08, respectively), which is consistent with Park et al. [48] who reported a stronger correlation between TNF-α and WC (r = 0.32), than between TNF-α and BMI (r = 0.24).

These findings suggest taking both pCML and pTNF-α into account as a metabolic parameter associated with the development of obesity, central obesity, or other NCDs. Knowledge of the impact of food processing on CML formation and the CML value in Indonesian foods could lead to new dietary recommendations for the general community, and especially for the people residing in the two Indonesian provinces that we studied.

## 5. Conclusions

A positive direct effect of dCML on pCML and of pTNF-α on WC was seen among Indonesian women. Dietary CML seems to have several paths to indirectly influence the increase in WC if compared to BMI. Thus, intake of CML-rich foods should be reduced to moderate amounts to avoid risk of central obesity in this population.

## Figures and Tables

**Figure 1 nutrients-11-03057-f001:**
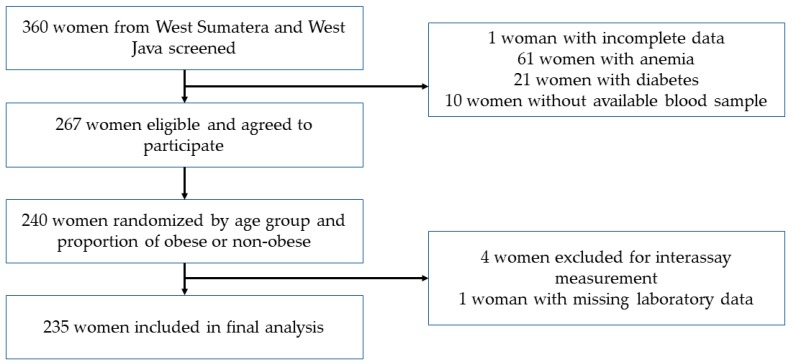
Flowchart of subject selection.

**Figure 2 nutrients-11-03057-f002:**
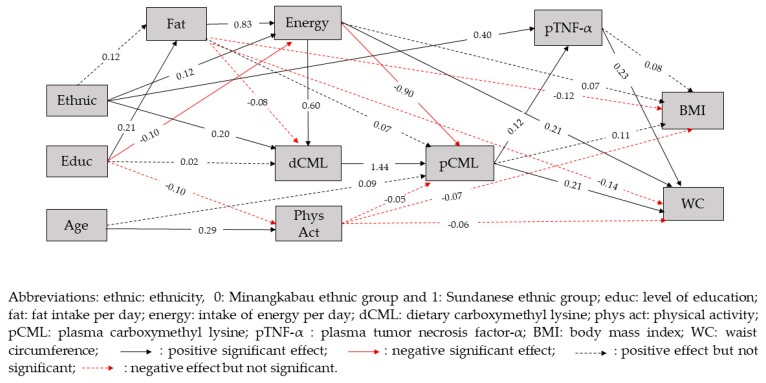
The fitted structural model with standardized parameter values.

**Table 1 nutrients-11-03057-t001:** General characteristics of non-obese, obese, non-central obesity and central obesity in Indonesian women living in selected areas of West Sumatra and West Java provinces.

Variable	Body Mass Index			Waist Circumference		
	Non-Obese (*n* = 121)	Obese (*n* = 114)	*p-*Value ^a^	Non-Central Obesity (*n* = 104)	Central Obesity (*n* = 131)	*p-*Value ^b^
Age (years)	36 (28–43) ^‡^	38 (31–44) ^‡^	0.541	36 (27–43) ^‡^ *	38 (32–44) ^‡^ *	0.031
Age group in years (*n*, %)						
19–40	78 (64.5)	72 (63.2)	0.942	70 (67.3)	80 (61.1)	0.394
41–50	43 (35.5)	42 (36.8)		34 (32.7)	51 (38.9)	
Education (*n*, %)						
Low	31 (25.6)	40 (35.1)	0.151	31 (29.8)	40 (30.5)	>0.999
Middle	90 (74.4)	72 (63.2)		73 (70.2)	91 (69.5)	
Ethnic group (*n,* %)						
Minangkabau	67 (55.4)	50 (43.9)	0.102	63 (60.6) ^*^	54 (41.2) *	0.005
Sundanese	54 (44.6)	64 (56.1)		41 (39.4)	77 (58.8)	
Physical activity (*n*, %)						
Inactive	85 (70.2)	89 (78.1)	0.172	74 (71.2)	100 (76.3)	0.368
Active	36 (29.8)	25 (21.9)		30 (28.8)	31 (23.7)	

Abbreviations: ^a^ The Mann-Whitney test was used to compare continuous data between women with non-obesity and women with obesity; and ^b^ between women with non-central obesity and women with central obesity. The Chi-square test was used to compare categorical data between the two groups. ^‡^: Data were non-normally distributed, presented as median (25th percentile–75th percentile). ***** Results were considered statistically significant at *p* < 0.05. Category of low education: not finished senior high school; middle: finished senior high school/has higher education. Category of inactive physical activity: <600 MET; active physical activity: ≥600 MET. Obese and central obesity were determined by body mass index ≥ 25 kg/m^2^ and waist circumference ≥ 80 cm, respectively.

**Table 2 nutrients-11-03057-t002:** Dietary CML, plasma CML and plasma TNF-α level among non-obese and obese or non-central obese and central obese Indonesian women living in selected areas of West Sumatra and West Java provinces (*n* = 235).

Variable	Body Mass Index			Waist Circumference		
	Non-Obese (*n* = 121)	Obese (*n* = 114)	*p-*Value ^a^	Non-Central Obese (*n* = 104)	Central Obese (*n* = 131)	*p-*Value ^b^
Dietary CML (mg/day)	2.1 (1.7–2.9) ^‡^	2.3 (1.7–3.2) ^‡^	0.206	2.1 (1.7–2.6) ^‡^ *	2.3 (1.7–3.4) ^‡^ *	0.040
Plasma CML (ng/mL)	22.3 (17.2–27.3) ^‡^	22.2 (17.3–30.2) ^‡^	0.177	21.0 (16.8–26.8) ^‡^ *	23.2 (18.4–30.1) ^‡^ *	0.026
TNF-α (IU/mL)	0.64 (0.51–0.92) ^‡^	0.72 (0.54–1.08) ^‡^	0.095	0.62 (0.48–0.77) ^‡^ **	0.76 (0.54–1.22) ^‡^ **	<0.001

Abbreviations: CML: carboxymethyl lysine; TNF-α: tumor necrosis factor-alpha. ^a^ Compared continuous data between non-obese and obese; and ^b^ between non-central and central obesity. ***** Mann-Whitney test statistically significant at *p* < 0.05; and ******
*p* < 0.001. ^‡^: Data presented as median (25th percentile–75th percentile).

**Table 3 nutrients-11-03057-t003:** Direct effects between research variables.

Effect	β (95% CI)	*p*-Value	Effect	β (95% CI)	*p-*Value
**Direct effect on WC**			**Direct effect on BMI**		
pTNF-α	0.23 (0.11, 0.35)	<0.001 **	pTNF-α	0.08 (−0.06, 0.20)	0.131
pCML	0.21 (0.09, 0.33)	0.001 *	pCML	0.11 (−0.02, 0.24)	0.046
Fat	−0.14 (−0.34, 0.07)	0.095	Fat	−0.12 (−0.34, 0.10)	0.136
Energy	0.21 (0.00, 0.41)	0.025 *	Energy	0.07 (−0.15, 0.29)	0.263
Phys Act	−0.06 (−0.18, 0.06)	0.155	Phys Act	−0.07 (−0.20, 0.05)	0.132
**Direct effect on pTNF-α**			**Direct effect on intake of fat**		
Ethnic	0.40 (0.28, 0.49)	<0.001 **	Educ	0.21 (0.08, 0.33)	0.001 **
pCML	0.12 (0.00, 0.24)	0.024 *	Ethnic	0.12 (−0.02, 0.24)	0.041
**Direct effect on pCML**			**Direct effect on intake of energy**		
dCML	1.44 (0.70, 3.23)	<0.001 **	Ethnic	0.12 (0.05, 0.19)	0.001 *
Energy	−0.90 (−2.21, −0.31)	0.001 *	Fat	0.83 (0.78, 0.87)	<0.001 **
Fat	0.07 (−0.27, 0.54)	0.347	Educ	−0.10 (−0.18, 0.03)	0.003 *
Age	0.09 (−0.05, 0.21)	0.100			
Phys Act	−0.05 (−0.18, 0.08)	0.238			
**Direct effect on dCML**			**Direct effect on physical activity**		
Ethnic	0.20 (0.10, 0.30)	<0.001 **	Age	0.29 (0.17, 0.40)	<0.001 **
Energy	0.60 (0.42, 0.77)	<0.001 **	Educ	−0.10 (−0.22, 0.02)	0.053
Educ	0.02 (−0.05, 0.10)	0.261			
Fat	−0.08 (−0.26, 0.11)	0.208			

Abbreviations: Ethnic: ethnicity; Educ: education; Fat: intake of fat per day; Energy: intake of energy per day; dCML: dietary carboxymethyl lysine; Phys Act: physical activity; pCML: plasma carboxymethyl lysine; pTNF-α: plasma tumor necrosis factor-α. ***** Results were considered statistically significant at <0.025 and ******
*p* < 0.001.

**Table 4 nutrients-11-03057-t004:** Indirect effects of independent variables on BMI and WC.

	Body Mass Index		Waist Circumference	
Effect	β (95%CI)	*p*-Value	β (95%CI)	*p*-Value
**Effects of ethnicity on BMI or WC**				
**Sum of indirect effects**	0.07 (0.01, 0.14)	0.014 *	0.18 (0.12, 0.25)	<0.001 **
**Specific indirect effect**				
via energy, dCML, pCML, and pTNF-α	0.00 (0.00, 0.01)	0.148	0.00 (0.00, 0.01)	0.025 *
via energy, dCML, and pCML	0.01 (0.00, 0.04)	0.046	0.02 (0.00, 0.07)	0.001 *
via energy, pCML, and pTNF-α	0.00 (−0.01, 0.00)	0.149	0.00 (−0.01, 0.00)	0.025
via energy and pCML	−0.01 (−0.05, 0.00)	0.047	−0.02 (−0.07, 0.00)	0.002 *
via energy	0.01 (−0.02, 0.04)	0.264	0.02 (0.00, 0.06)	0.025
via dCML, pCML, and pTNF-α	0.00 (0.00, 0.01)	0.148	0.01 (0.00, 0.02)	0.024 *
via dCML and pCML	0.03 (−0.01, 0,08)	0.046	0.06 (0.02, 0.11)	0.001 *
via pTNF-α	0.03 (−0.02, 0.08)	0.131	0.09 (0.04, 0.15)	<0.001 **
**Effects of education level on BMI or WC**				
**Sum of indirect effects**	0.00 (0.02, 0.03)	0.343	0.01 (−0.01, 0.05)	0.168
**Specific indirect effect**				
via fat, energy, dCML, pCML, and pTNF-α	0.00 (0.00, 0.01)	0.149	0.00 (0.00, 0.02)	0.025
via fat, energy, dCML, and pCML	0.02 (0.00, 0.06)	0.046	0.03 (0.01, 0.10)	0.001 *
via fat, energy, pCML, and pTNF-α	0.00 (−0.01, 0.00)	0.149	0.00 (−0.02, 0.00)	0.025
via fat, energy, and pCML	−0.02 (−0.07, 0.00)	0.047	−0.03 (−0.10, 0.01)	0.002 *
via fat and energy	0.01 (−0.03, 0.24)	0.278	0.03 (0.00, 0.09)	0.025
via energy, dCML, pCML, and pTNF-α	0.00 (0.00, 0.00)	0.150	0.00 (−0.01, 0.00)	0.027
via energy, dCML, and pCML	−0.01 (−0.04, 0.00)	0.049	−0.02 (−0.06, 0.00)	0.004 *
via energy, pCML and pTNF-α	0.00 (0.00, 0.00)	0.151	0.00 (0.00, 0.01)	0.028
via energy and pCML	0.01 (0.00, 0.04)	0.049	0.02 (0.00, 0.06)	0.004 *
via energy	−0.01 (−0.04, 0.02)	0.265	−0.02 (−0.05, 0.00)	0.028
**Effects of fat intake on BMI or WC**				
**Total effects**				
**Sum of indirect effects**	0.06 (−0.13, 0.238)	0.278	0.17 (−0.01, 0.34)	0.034
**Specific indirect effect**				
via energy, dCML, pCML, and pTNF-α	0.01 (−0.01, 0.03)	0.148	0.02 (0.00, 0.07)	0.024 *
via energy, dCML, and pCML,	0.08 (−0.01, 0.27)	0.046	0.14 (0.04, 0.41)	0.001 *
via energy, pCML, and pTNF-α	−0.01 (−0.03, 0.01)	0.148	−0.02 (−0.07, 0.00)	0.025 *
via energy and pCML	−0.08 (−0.28, 0.01)	0.046	−0.15 (−0.42, −0.04)	0.001 *
via energy	0.06 (−0.124, 0.24)	0.263	0.17 (0.00, 0.34)	0.025 *
**Effects of energy intake on BMI or WC**				
**Sum of indirect effects**	0.00 (−0.04, 0.03)	0.395	−0.01 (−0.07, 0.05)	0.388
**Specific indirect effect**				
via dCML, pCML, and pTNF-α	0.01 (−0.01, 0.04)	0.148	0.02 (0.00, 0.08)	0.022 *
via dCML and pCML	0.09 (−0.02, 0.33)	0.046	0.17 (0.05, 0.49)	0.001 *
via pCML and pTNF-α	−0.01 (−0.04, 0.01)	0.148	−0.02 (−0.09, 0.00)	0.025 *
via pCML	−0.10 (−0.34, 0.02)	0.046	−0.18 (−0.51, −0.05)	0.001 *
**Effects of dCML on BMI or WC**				
**Sum of indirect effects**	0.17 (−0.01, 0.52)	0.032	0.33 (0.12, 0.83)	<0.001 **
**Specific indirect effect**				
via pCML, and pTNF-α	0.01 (−0.01, 0.06)	0.148	0.04 (0.00, 0.13)	0.024 *
via pCML	0.16 (−0.03, 0.50)	0.046	0.29 (0.10, 0.75)	0.001 *
**Effects of pCML on BMI or WC** **Sum of indirect effects**				
via pTNF-α	0.01 (−0.01, 0.03)	0.148	0.03 (0.00, 0.07)	0.024 *

Abbreviations: Ethnic: ethnicity (0: Minangkabau ethnic group and 1: Sundanese ethnic group); Educ: level of education; fat: intake of fat per day; energy: intake of energy per day; dCML: dietary carboxymethyl lysine; pCML: plasma carboxymethyl lysine; pTNF-α: plasma tumor necrosis factor-α. ***** Results were considered statistically significant at <0.025 and ******
*p* < 0.001.

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
