# Peer review of "Dietary and Plasma Carboxymethyl Lysine and Tumor Necrosis Factor-α as Mediators of Body Mass Index and Waist Circumference among Women in Indonesia"

_nutrients, 2019, doi:10.3390/nu11123057_

Round 1

Reviewer 1 Report

Liman and colleagues investigated the mediator effects of dCML, pCML and pTNF-α on body mass index (BMI) and waist circumference (WC) among Indonesian women. They performed a cross-sectional study among 235 non-diabetic, non-anemic, and non-smoking women aged 19-50 years from selected mountainous and coastal areas of West Sumatera and West Java.

This is an interesting read and addresses a field of emerging importance: the obesity in Indonesian women.

My main concern, however, is that this is not a novel concept, however few information are available about Indonesian women.

Additional suggestions are listed below:

Regarding introduction – could be shortened.

Regarding discussion. Some observations are missing. Some paper described the women rarely follow such a lifestyle, and this is strongly influenced by their income level, social role, education and culture [Sciomer at al doi: 10.1177/2047487318810560.]. Need to be discuss in discussion section

Result section line 208-210 This is incomprehensible

Author Response

Response to Reviewer 1 Comment

Dear Reviewer 2,

We would like to thank you for your careful and detailed reading of the manuscript. The manuscript has been revised accordingly to your comments.

Point 1: Liman and colleagues investigated the mediator effects of dCML, pCML and pTNF-α on body mass index (BMI) and waist circumference (WC) among Indonesian women. They performed a cross-sectional study among 235 non-diabetic, non-anemic, and non-smoking women aged 19-50 years from selected mountainous and coastal areas of West Sumatera and West Java.

 This is an interesting read and addresses a field of emerging importance: the obesity in Indonesian women.

My main concern, however, is that this is not a novel concept, however few information are available about Indonesian women.

Response 1: We thank the Reviewer for the comment.

Point 2: Regarding introduction – could be shortened

 Response 2: We have made changes to the introduction. Hopefully the changes meet your criteria.

Point 3: Regarding discussion. Some observations are missing. Some paper described the women rarely follow such a lifestyle, and this is strongly influenced by their income level, social role, education and culture [Sciomer at al doi: 10.1177/2047487318810560.]. Need to be discuss in discussion section.

 Response 3: We thank the Reviewer for the suggestion. We have made additions to the Discussion, in accordance with the suggestion of Reviewer 1, namely in line 366-380, which we have highlighted in yellow.

Point 4: Result section line 208-210 This is incomprehensible.

 Response 4: We thank the Reviewer for the comment. the authors wished to indicate that ethnicity influences the risk of obesity and central obesity, although statistically the difference in prevalence was significant only between non-central obesity and central obesity.

We have changed the sentence to “Sundanese women had a higher prevalence of obesity and central obesity than Minangkabau women, the difference being statistically significant for central obesity.”

 Reviewer 2 Report

The manuscript is comprehensive and adequately addresses its objective, the few corrections and comments in the text should be considered and incorporated for the paper.

Author Response

Response to Reviewer 2 Comment

Dear Reviewer 2,

We would like to thank you for your detailed reading and constructive comment. The manuscript has been revised accordingly to your comment.

Point 1: This sentence “The present study demonstrated that pTNF-α has a stronger association with WC than with BMI (β=023 and β=0.08, respectively), that is inconsistent with Park et al. 6 who reported a stronger correlation between TNF-α and WC (r=0.32), than between TNF-α and BMI (r=0.24). “ (371-373) is not clear since both results are in agreement.

Response 1: We thank the Reviewer for the comment. Yes, we made a mistake here. We have changed “inconsistent” to “consistent” in line 420 (highlighted in yellow).
